# Surgical Trends and Complications in Partial and Radical Nephrectomy: Results from the GRAND Study

**DOI:** 10.3390/cancers16010097

**Published:** 2023-12-24

**Authors:** Nikolaos Pyrgidis, Gerald Bastian Schulz, Christian Stief, Iulia Blajan, Troya Ivanova, Annabel Graser, Michael Staehler

**Affiliations:** Department of Urology, LMU University Hospital, LMU Munich, 81377 Munich, Germany; nikolaos.pyrgidis@med.uni-muenchen.de (N.P.); gerald.schulz@med.uni-muenchen.de (G.B.S.); christian.stief@med.uni-muenchen.de (C.S.); iulia.blajan@med.uni-muenchen.de (I.B.); troya.ivanova@med.uni-muenchen.de (T.I.); annabel.graser@med.uni-muenchen.de (A.G.)

**Keywords:** cohort study, partial nephrectomy, radical nephrectomy, perioperative outcomes, mortality

## Abstract

**Simple Summary:**

Studies about the current trends in renal cancer surgery and its perioperative outcomes are lacking. Using the nationwide data of Germany from 2005 to 2021, we found that the utilization of partial nephrectomy substantially increased, while the utilization of radical nephrectomy substantially decreased in the last years. Patients selected for radical nephrectomy had more comorbidities and risk factors compared to patients selected for partial nephrectomy. Our analyses suggest that patients undergoing radical nephrectomy present worse perioperative morbidity and mortality, as well as prolonged hospitalization, compared to patients undergoing partial nephrectomy.

**Abstract:**

Background: We aimed to evaluate the current trends in renal cancer surgery, as well as to compare the perioperative outcomes of partial versus radical nephrectomy. Methods: We used the GeRmAn Nationwide inpatient Data (GRAND), provided by the Research Data Center of the Federal Bureau of Statistics (2005–2021). We report the largest study in the field, with 317,843 patients and multiple patient-level analyses. Results: Overall, 123,924 (39%) patients underwent partial and 193,919 (61%) underwent radical nephrectomy in Germany from 2005 to 2021. Of them, 57,308 (18%) were operated on in low-, 142,702 (45%) in intermediate-, and 117,833 (37%) in high-volume centers. A total of 249,333 (78%) patients underwent open, 44,994 (14%) laparoscopic, and 23,516 (8%) robotic nephrectomy. The number of patients undergoing renal surgery remained relatively stable from 2005 to 2021. Over the study period, the utilization of partial nephrectomy increased threefold, while radical nephrectomy decreased by about 40%. After adjusting for major risk factors in the multivariate regression analysis, radical nephrectomy was associated with 3.2-fold higher odds (95% CI: 3.2 to 3.9, *p* < 0.001) of 30-day mortality, longer hospitalization by 1.9 days (95% CI: 1.9 to 2, *p* < 0.001), and higher inpatient costs by EUR 1778 (95% CI: 1694 to 1862, *p* < 0.001) compared to partial nephrectomy. Furthermore, radical nephrectomy had a higher risk of in-hospital transfusion (*p* < 0.001), sepsis (*p* < 0.001), acute respiratory failure (*p* < 0.001), acute kidney disease (*p* < 0.001), acute thromboembolism (*p* < 0.001), surgical wound infection (*p* < 0.001), ileus (*p* < 0.001), intensive care unit admission (*p* < 0.001), and pancreatitis (*p* < 0.001). Conclusions: More patients are offered partial nephrectomy in Germany. Patients undergoing radical nephrectomy present with a higher rate of concomitant risk factors and have increased perioperative morbidity and mortality, prolonged hospitalization, and increased in-hospital costs.

## 1. Introduction

Renal cancer accounts for over 3% of all new cancer cases worldwide, affecting more than 430,000 individuals every year [1]. The increasing prevalence of risk factors for renal cancer, such as obesity, hypertension, and chronic renal disease, has contributed to a rising incidence of renal cancer globally [2]. Accordingly, the recent technological improvements, in combination with the wide implementation of cross-sectional imaging, have led to earlier diagnosis of renal cancer in many patients [3]. Therefore, 75% of all newly diagnosed renal masses are asymptomatic, incidental findings and smaller than 7 cm in diameter. About 80% of all surgically resected renal tumors are malignant in the final histology [4].

Despite the recent advancements in other local or systemic therapies, partial nephrectomy is considered the standard treatment for patients with suspected renal cancer, if technically feasible [5,6]. Partial nephrectomy has supplanted radical nephrectomy as the preferred treatment modality, given that it is associated with superior functional and equivalent oncological outcomes [7]. Radical nephrectomy remains the treatment of choice for tumors in which partial nephrectomy is not possible [8]. Both partial and radical nephrectomy can be performed with either an open or a minimally invasive approach [9]. Nevertheless, nephrectomy is associated with perioperative mortality and morbidity, as well as with prolonged hospital stay, intensive care unit admission, and significant treatment-related costs [10,11].

Currently, there is a global trend toward the centralization of complex urological operations in healthcare systems [12]. This shift is supported by the accumulating evidence indicating that increased annual hospital volume leads to improved perioperative outcomes for major urological operations. [13]. Therefore, the EAU Guideline Panel on Renal Cell Carcinoma recommends that hospitals should annually perform at least forty partial nephrectomies [14]. Nevertheless, this recommendation is based on a low level of evidence derived from retrospective studies with relatively low numbers of included patients. Indeed, there is a paucity of existing studies attempting to identify a hospital volume threshold for annual kidney cancer surgery cases that may improve perioperative outcomes. Similarly, studies assessing the perioperative complications in patients undergoing kidney cancer surgery are lacking. In this scope, we aimed to evaluate the current trends in partial and radical nephrectomy, and to compare the perioperative outcomes of partial versus radical nephrectomy through the largest study in the field.

## 2. Methods

### 2.1. GeRmAn Nationwide Inpatient Data (GRAND)

For the present analysis, we used the GeRmAn Nationwide inpatient Data (GRAND). The GRAND study contains all German inpatient data from 2005 to 2021 apart from military, psychiatric, and forensic cases. These data are stored in an anonymized format at the Research Data Center of the German Bureau of Statistics, and they were retrieved for further analysis upon agreement (LMU—4710-2022). To ensure anonymity, the Research Data Center excludes patient groups with fewer than three baseline characteristics or inpatient complications and does not allow hospital-level comparisons for any outcome. After the 2004 implementation of a diagnosis- and procedure-related remuneration system in Germany (German diagnosis-related groups (DRGs)), all hospitals need to transfer the in-hospital patient data (e.g., coexisting conditions, surgical procedures, perioperative outcomes) to the Institute for the Hospital Remuneration System to receive their remuneration. These patient data are coded based on the International Statistical Classification of Diseases and Related Health Problems, 10th revision, German modification (ICD-10-GM) and the German Procedure Classification (OPS).

### 2.2. Selection Criteria

We included all patients undergoing radical (OPS code: 5-554.4, 5-554.a) and partial nephrectomy (OPS code: 5-553) for suspected renal tumors. To obtain patient data on procedures, concurrent conditions, and inpatient complications, we used the available diagnostic and procedural codes (ICD-10-GM and OPS). The primary outcome of the present analysis was to assess surgical trends in patients undergoing radical or partial nephrectomy. Secondary outcomes included the effects of radical versus partial nephrectomy on 30-day mortality and perioperative complications (i.e., transfusion, sepsis, acute respiratory failure, acute kidney disease, acute thromboembolism, surgical wound infection, ileus, intensive care unit admission, and pancreatitis). We also analyzed hospital revenues and length of hospital stay. Moreover, we compared 30-day mortality and perioperative complications in patients undergoing radical versus partial nephrectomy separately in high-, intermediate-, and low-volume centers. Given that there is no consensus on the definition of high-volume centers, we defined high-volume centers as those that perform at least 100 nephrectomies (partial and radical) per year. Similarly, intermediate-volume centers were defined as those that perform between 40 and 99 nephrectomies per year, and low-volume centers were defined as those that perform less than 40 nephrectomies per year.

### 2.3. Data Synthesis and Statistical Analysis

Our research team did not have direct access to patient-level data. Thus, all statistical analyses were performed on our behalf by the Research Data Center of the German Bureau of Statistics, based on R codes developed by our research team (source: Research Data Center of the Federal Bureau of Statistics, DRG Statistics 2005–2021; own calculations). Subsequently, the summary results were provided to our research group for further evaluation. Approval by an ethics committee or informed patient consent was not required based on the German legislation.

All hospitals performing renal surgeries were identified through their postal codes and were further subclassified based on their annual caseload as low-volume centers (<40 cases/year), intermediate-volume centers fulfilling the EAU recommendation (40–99 cases/year), and high-volume centers (≥100 cases/year). The corresponding comparisons among low- (<40 cases/year), intermediate- (40–99 cases/year), and high-volume centers (≥100 cases/year) were performed with the chi-squared test and the Kruskal–Wallis test. Accordingly, all comparisons between patients undergoing partial versus radical nephrectomy were performed with the chi-squared test and the Mann–Whitney U test. All continuous variables were calculated as medians with interquartile ranges (IQRs), and all categorical variables were calculated as frequencies with proportions.

We conducted multiple multivariable logistic and linear regression analyses to evaluate the effect of the type of surgery and the effect of the annual hospital caseload on inpatient outcomes (i.e., 30-day mortality, perioperative complications, length of hospital stay, and hospital revenues). All regression models were adjusted for sex, age, obesity, history of chronic obstructive pulmonary disease, chronic heart failure, chronic kidney disease, cerebrovascular accident, hypertension, and diabetes, as well as for the surgical approach and the year of operation. Odds ratios (ORs) with 95% confidence intervals (CIs) were estimated for all logistic models, and two-sided *p*-values lower than 0.05 were considered statistically significant. The log-rank test with Kaplan–Meier analyses was used to assess the effects of radical versus partial nephrectomy on 30-day mortality.

## 3. Results

### 3.1. Baseline Characteristics

A total of 317,843 patients with a median age of 66 years (IQR: 56–74) underwent kidney cancer surgery, with 188,123 (59%) being male; 179,386 (56%) had hypertension, 65,605 (21%) has chronic kidney disease, and 57,155 (18%) had diabetes. Overall, 123,924 (39%) patients underwent partial and 193,919 (61%) underwent radical nephrectomy. A total of 249,333 (78%) patients underwent open, 44,994 (14%) laparoscopic, and 23,516 (7.4%) robotic nephrectomy. Based on the annual hospital caseload volume for kidney cancer surgery, 57,308 (18%) operations (17,108 (30%) partial and 40,200 (70%) radical nephrectomies) were performed in low-volume centers, 142,702 (45%) operations (52,966 (37%) partial and 89,736 (63%) radical nephrectomies) in intermediate-volume centers, and 117,833 (37%) operations (53,850 (46%) partial and 63,983 (54%) radical nephrectomies) in high-volume centers.

Patients undergoing radical nephrectomy were older (*p* < 0.001) and had higher proportions of diabetes (*p* < 0.001), chronic heart failure (*p* < 0.001), chronic obstructive pulmonary disease (*p* < 0.001), chronic kidney disease (*p* < 0.001), cerebrovascular disease (*p* < 0.001), and dementia (*p* < 0.001) compared to patients undergoing partial nephrectomy. Laparoscopic or robotic surgical approaches were preferred more often in patients undergoing partial nephrectomy (*p* < 0.001). The latter was also observed in the separate analyses for low-, intermediate-, and high-volume centers. The baseline characteristics of all patients undergoing renal surgery are presented in Table 1, and the corresponding baseline characteristics of patients operated on in low-, intermediate-, and high-volume centers are presented in the Appendix A.

The number of patients undergoing renal surgery increased moderately from 17,360 cases in 2005 to 18,686 in 2021. Interestingly, the number of patients undergoing partial nephrectomy substantially increased throughout these years, from 3358 cases in 2005 to 10,153 in 2021, while the number of patients undergoing radical nephrectomy substantially decreased throughout the same period, from 14,002 cases in 2005 to 8533 in 2021. The latter was also observed in low-, intermediate-, and high-volume centers. Nevertheless, the increase in partial nephrectomies and the decrease in radical nephrectomies were steeper in high-volume centers. In particular, in high-volume centers, 1427 (2.6%) partial nephrectomies and 4634 (7.2%) radical nephrectomies were performed in 2005, compared to 4485 (8.3%) partial nephrectomies and 2857 (4.5%) radical nephrectomies in 2021. The number of patients undergoing renal surgery was not affected during the COVID-19 pandemic. The annual trends for radical and partial nephrectomy are depicted in Figure 1, whereas the corresponding annual trends for radical and partial nephrectomy in low-, intermediate-, and high-volume centers are depicted in the Appendix A.

### 3.2. Effects of Renal Surgery on Perioperative Morbidity, Mortality, Hospital Stay, and Costs

Overall, radical nephrectomy, compared to partial nephrectomy, was significantly associated with higher odds of transfusion (25% versus 12%; OR: 2, 95% CI: 1.9 to 2, *p* < 0.001), sepsis (3.1% versus 1%; OR: 2.6, 95% CI: 2.4 to 2.8, *p* < 0.001), acute respiratory failure (5.4% versus 3.6%; OR: 1.6, 95% CI: 1.5 to 1.7, *p* < 0.001), acute kidney disease (5.9% versus 4%; OR: 1.6, 95% CI: 1.5 to 1.7), acute thromboembolism (0.9% versus 0.5%; OR: 1.9, 95% CI: 1.7 to 2, *p* < 0.001), surgical wound infection (0.7% versus 0.3%; OR: 2, 95% CI: 1.8 to 2.2, *p* < 0.001), ileus (2% versus 1.3%; OR: 1.4, 95% CI: 1.3 to 1.5, *p* < 0.001), intensive care unit admission (21% versus 16%; OR: 1.2, 95% CI: 1.2 to 1.2, *p* < 0.001), and pancreatitis (0.3% versus 0.1%; OR: 2.1, 95% CI: 1.8 to 2.5, *p* < 0.001). Similarly, radical nephrectomy was associated with longer hospital stay by 1.9 days (95% CI: 1.9 to 2, *p* < 0.001) and higher inpatient costs by EUR 1778 (95% CI: 1694 to 1862, *p* < 0.001). The multivariable analysis is presented in Table 2 and Table 3. In the separate multivariable analyses of patients undergoing renal surgery in low-, intermediate-, and high-volume centers, radical nephrectomy was associated with worse perioperative outcomes compared to partial nephrectomy. The corresponding findings are presented in the Appendix A.

A total of 4202 (1.3%) in-hospital deaths were observed within 30 days of surgery. Of these, 540 (0.4%) deaths occurred among patients undergoing partial nephrectomy and 3662 (1.9%) among patients undergoing radical nephrectomy. The numbers of 30-day in-hospital deaths after partial nephrectomy were 116 (0.7%) in low-volume centers, 234 (0.4%) in intermediate-volume centers, and 190 (0.4%) in high-volume centers. Meanwhile, the numbers of 30-day in-hospital deaths after radical nephrectomy were 917 (2.3%) in low-volume centers, 1704 (1.9%) in intermediate-volume centers, and 1041 (1.6%) in high-volume centers.

Radical nephrectomy was associated with 3.5-fold higher odds (95% CI: 3.2 to 3.9, *p* < 0.001) of 30-day mortality compared to partial nephrectomy in the whole study population. In low-volume centers, radical nephrectomy was associated with 2.8-fold higher odds (95% CI: 2.3 to 3.5, *p* < 0.001) of 30-day mortality compared to partial nephrectomy. In intermediate-volume centers, radical nephrectomy was associated with 3.6-fold higher odds (95% CI: 3.1 to 4.1, *p* < 0.001) of 30-day mortality compared to partial nephrectomy. In high-volume centers, radical nephrectomy was associated with 3.7-fold higher odds (95% CI: 3.2 to 4.4, *p* < 0.001) of 30-day mortality compared to partial nephrectomy. The latter was also observed in the time-to-death analysis for the whole study population, as well as for low-, intermediate-, and high-volume centers (log-rank test for all comparisons: *p* < 0.001). The corresponding Kaplan–Meier analyses are presented in Figure 2 and Appendix A.

## 4. Discussion

The present high-volume study suggests that the annual cases of partial nephrectomy have undergone a threefold increase, while the annual cases of radical nephrectomy have undergone an important decrease of 40% in the last few years. Interestingly, the number of patients undergoing renal surgery has remained largely unchanged from 2005 to 2021. In high-volume centers, the proportion of partial to radical nephrectomies is higher compared to in intermediate- and low-volume centers. Furthermore, it seems that minimally invasive renal surgery is preferred in only a small proportion of patients requiring nephrectomy in Germany. It should be noted that patients undergoing radical nephrectomy displayed worse baseline characteristics compared to those undergoing partial nephrectomy. Nevertheless, after adjusting for these characteristics, we found that radical nephrectomy was associated with 195% higher odds of surgical wound infection, 160% higher odds of sepsis, 110% higher odds of pancreatitis, 100% higher odds of transfusion, 90% higher odds of acute thromboembolism, 60% higher odds of acute respiratory failure and acute kidney disease, 40% higher odds of ileus, and 20% higher odds of intensive care unit admission. Similarly, radical nephrectomy, compared to partial nephrectomy, was associated with 250% higher odds of 30-day in-hospital mortality, as well as with longer hospital stay and higher inpatient costs. The latter was also observed in the separate analysis between partial and radical nephrectomy in low-, intermediate-, and high-volume centers.

Our findings demonstrate that the utilization of partial nephrectomy has considerably increased in recent years. The latter might be predominantly attributed to the fact that current major guidelines recommend partial nephrectomy over radical nephrectomy whenever possible [14,15,16]. Studies from the US and the UK are in line with our findings demonstrating that the use of partial nephrectomy has increased significantly over time for both small and larger renal masses [17,18,19]. A large prospective study from Singapore suggested a shift towards nephron-sparing surgery for clinically localized tumors [20]. Accordingly, a nationwide study from the Netherlands demonstrated a clear increase in nephron-sparing management either with active surveillance, partial nephrectomy, or focal therapy over time for cT1a tumors. Conversely, for cT1b tumors, radical nephrectomy remained the most common treatment modality, although patients in high-volume centers more often underwent partial nephrectomy [21].

Still, despite the advancements in surgical techniques for nephrectomy, it seems that, in Germany, minimally invasive radical or partial nephrectomy is performed in a relatively small amount of patients compared to the US [22,23]. Even though the number of suspicious renal masses diagnosed every year in Germany has increased [24], the overall number of nephrectomy cases that are performed every year has remained stable in recent years [25,26]. The latter might be explained by the fact that an increasing number of patients with diagnosed renal masses undergo active surveillance or other ablative techniques [27,28,29,30]. Moreover, the fact that partial nephrectomy cases have increased by 300% while radical nephrectomy cases have decreased by 40% over the years might indicate that renal cancer is often diagnosed at less-advanced tumor stages. Indeed, large epidemiological studies suggest that the incidence of renal cancer continues to rise, mainly for early-stage tumors, whereas that of advanced stages has declined [31,32].

It should be highlighted that renal cancer surgery is associated with low in-hospital mortality and morbidity, which are higher in patients undergoing radical nephrectomy. Large prospective comparative studies indicate that in patients with similar renal masses in terms of diameter and location, partial and radical nephrectomy present comparable perioperative outcomes [8,33,34]. Still, in the present analysis, we could not perform an adjustment between the two groups in terms of their tumor characteristics. Based on the previous notion, patients undergoing radical nephrectomy presented worse baseline parameters and worse tumor characteristics, which might be the cause of the observed higher morbidity and mortality, as well as of the prolonged hospital stay and the increased costs compared to partial nephrectomy. It should be noted that patients with larger or more advanced tumors are more likely to require radical nephrectomy. The latter introduces a selection bias in the present analysis, since the inherent differences in tumor characteristics, the stage of the tumor, and patients’ general condition may impact the estimated differences in terms of outcomes when comparing partial nephrectomy and radical nephrectomy.

In the absence of detailed tumor characteristics, it is difficult to determine how an advanced tumor stage might affect outcomes. In an attempt to overcome this selection bias, we adjusted for multiple important risk factors in the multivariate regression analysis. However, without specific tumor characteristics, there may be major residual confounding due to unmeasured variables related to tumor stage and extension. Nevertheless, it should be stressed that radical nephrectomy was associated with worse perioperative outcomes for all estimates in low-, intermediate-, and high-volume centers compared to partial nephrectomy.

As this report, to the best of our knowledge, is the largest study on trends and perioperative outcomes in renal cancer surgery, our data have limitations that need to be considered. First of all, our analyses were derived from retrospective administrative data and are prone to coding errors and misclassifications. Although these administrative data present a high degree of accuracy and are regularly evaluated by independent physician task forces from healthcare insurance companies, important information on renal cancer surgery is not collected. In particular, the tumor size and location, the patient’s laboratory findings, the performance of a prior renal biopsy, the operative time, and the oncological status (i.e., histology findings, TNM classification, and surgical margins) are not available in the GRAND study. Similarly, data on mortality and morbidity after hospital discharge, readmission rates and causes of reoperation, functional outcomes, and follow-up data were not collected in the GRAND study. Moreover, the GRAND study does not provide information on the decision-making process between partial and radical nephrectomy, including patient preferences. Furthermore, the degree of baseline characteristics or perioperative complications such as chronic kidney disease, sepsis, or other perioperative complications cannot be retrieved. It was also beyond the scope of the present study to assess the number of operations performed in urological versus non-urological surgical departments (e.g., general surgery, pediatric surgery). Similarly, we did not consider exploring differences in the perioperative outcomes across different patient groups (such as younger patients, patients with obesity, or those with different levels of chronic kidney disease). Finally, our analyses are restricted to data for Germany and, thus, cannot be extrapolated to other healthcare systems. Accordingly, our findings may have limited generalizability, especially if the patient population undergoing radical nephrectomy differs systematically from those undergoing partial nephrectomy in terms of tumor characteristics and patients’ general condition. Nevertheless, in an attempt to overcome these limitations, our holistic and critical approach, combined with the size and nature of the GRAND study, leads to solid conclusions.

## 5. Conclusions

The present high-volume, nationwide, real-world data from Germany demonstrate an increased utilization of partial nephrectomy in renal cancer surgery. In recent years, the annual cases of partial nephrectomy have exceeded those of radical nephrectomy. Only one-fourth of all patients undergoing renal surgery are treated with a minimally invasive surgical approach. Based on our findings, patients undergoing radical nephrectomy present with worse baseline characteristics and experience higher perioperative morbidity and mortality, prolonged length of hospital stay, and increased in-hospital costs compared to patients undergoing partial nephrectomy. Still, this study’s conclusions about the superiority of partial nephrectomy in terms of perioperative outcomes should be interpreted with caution, as the decision to perform radical nephrectomy might be driven by clinical factors related to an advanced tumor stage or tumor location and/or anatomical complexity.

## Figures and Tables

**Figure 1 cancers-16-00097-f001:**
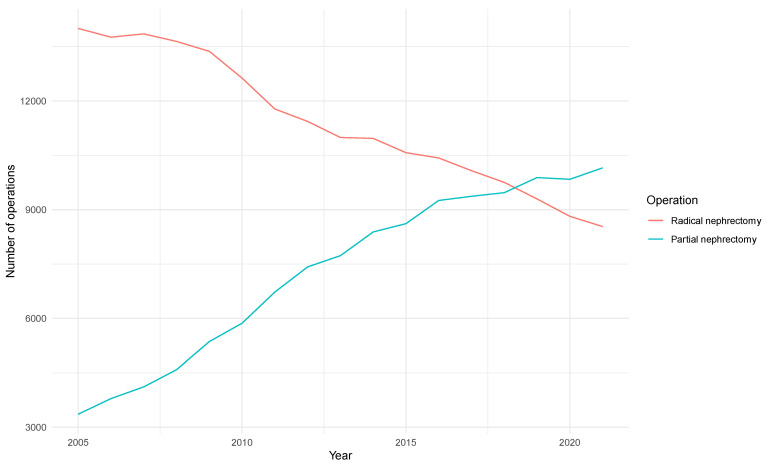
The annual trends for radical and partial nephrectomy.

**Figure 2 cancers-16-00097-f002:**
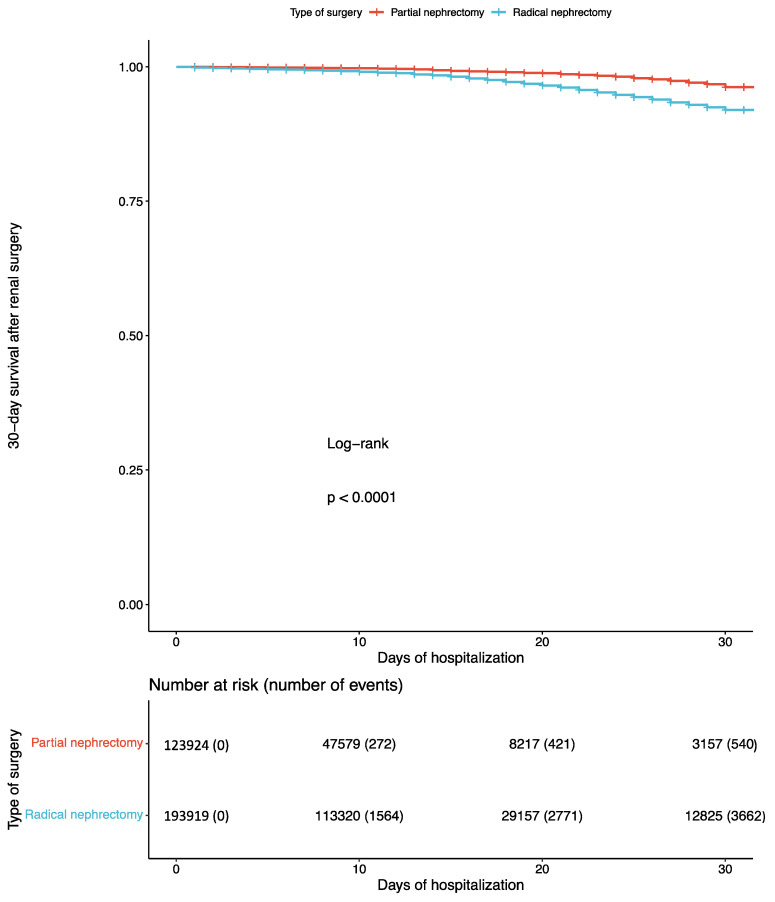
Kaplan–Maier curve for the 30-day survival in patients undergoing radical versus partial nephrectomy.

**Table 1 cancers-16-00097-t001:** Baseline characteristics of the included patients based on the type of renal cancer surgery: Variables are presented as medians with interquartile ranges or as frequencies with proportions. The Mann–Whitney test was performed for comparisons between continuous variables, and the chi-squared test was used for categorical variables. The bold cells indicate statistically significant *p*-values.

Characteristic	Overall, n = 317,843	Partial Nephrectomy, n = 123,924	Radical Nephrectomy, n = 193,919	*p*-Value
**Age (years)**	66 (56–74)	65 (56–73)	67 (56–75)	**<0.001**
**Males**	188,123 (59%)	77,117 (62%)	111,006 (57%)	**<0.001**
**Diabetes**	57,155 (18%)	21,319 (17%)	35,836 (18%)	**<0.001**
**Chronic heart failure**	20,140 (6.3%)	5722 (4.6%)	14,418 (7.4%)	**<0.001**
**Chronic obstructive pulmonary disease**	23,142 (7.3%)	8694 (7%)	14,448 (7.5%)	**<0.001**
**Chronic kidney disease**	65,605 (21%)	16,380 (13%)	49,225 (25%)	**<0.001**
**Cerebrovascular disease**	7607 (2.4%)	2126 (1.7%)	5481 (2.8%)	**<0.001**
**Dementia**	3689 (1.2%)	693 (0.6%)	2996 (1.5%)	**<0.001**
**Hypertension**	179,386 (56%)	70,382 (57%)	109,004 (56%)	**0.001**
**Obesity**	32,008 (10%)	12,384 (10%)	19,624 (10%)	0.25
**Operative technique**				**<0.001**
Open	249,333 (78%)	89,227 (72%)	160,106 (83%)	
Laparoscopic	44,994 (14%)	15,528 (13%)	29,466 (15%)	
Robotic	23,516 (7.4%)	19,169 (15%)	4347 (2.2%)	

**Table 2 cancers-16-00097-t002:** Multivariable logistic regression analysis for the effects of the type of surgery on transfusion, sepsis, acute respiratory failure, acute kidney disease, acute thromboembolism, and surgical wound infection. All models are adjusted for sex, age, obesity, history of chronic obstructive pulmonary disease, chronic heart failure, chronic kidney disease, cerebrovascular accident, hypertension, diabetes, surgical approach, and year of operation. The bold cells indicate statistically significant *p*-values. CI: confidence interval, OR: odds ratio.

Complications		Partial Nephrectomy	Radical Nephrectomy
**Transfusion**	**Events**	15,315 (12%)	49,169 (25%)
	**OR (95% CI)**	-	2 (1.9, 2)
	** *p* ** **-Value**	-	**<0.001**
**Sepsis**	**Events**	1260 (1%)	6018 (3.1%)
	**OR (95% CI)**	-	2.6 (2.4, 2.8)
	** *p* ** **-Value**	-	**<0.001**
**Acute respiratory failure**	**Events**	4448 (3.6%)	10,438 (5.4%)
	**OR (95% CI)**	-	1.6 (1.5, 1.7)
	** *p* ** **-Value**	-	**<0.001**
**Acute kidney disease**	**Events**	4985 (4%)	11,408 (5.9%)
	**OR (95% CI)**	-	1.6 (1.5, 1.7)
	** *p* ** **-Value**	-	**<0.001**
**Acute thromboembolism**	**Events**	630 (0.5%)	1802 (0.9%)
	**OR (95% CI)**	-	1.9 (1.7, 2)
	** *p* ** **-Value**	-	**<0.001**
**Surgical wound infection**	**Events**	391 (0.3%)	1369 (0.7%)
	**OR (95% CI)**	-	2 (1.8, 2.2)
	** *p* ** **-Value**	-	**<0.001**

**Table 3 cancers-16-00097-t003:** Multivariable linear and logistic regression analysis for the effects of the type of surgery on ileus, 30-day mortality, ICU admission, length of hospital stay, costs, and pancreatitis. All models are adjusted for sex, age, obesity, history of chronic obstructive pulmonary disease, chronic heart failure, chronic kidney disease, cerebrovascular accident, hypertension, diabetes, surgical approach, and year of operation. The bold cells indicate statistically significant *p*-values. CI: confidence interval, ICU: intensive care unit, OR: odds ratio.

Complications		Partial Nephrectomy	Radical Nephrectomy
**Ileus**	**Events**	1623 (1.3%)	3903 (2%)
	**OR (95% CI)**	-	1.4 (1.3, 1.5)
	***p*-Value**	-	**<0.001**
**30-Day mortality**	**Events**	540 (0.4%)	3662 (1.9%)
	**OR (95% CI)**	-	3.5 (3.2, 3.9)
	***p*-Value**	-	**<0.001**
**ICU admission**	**Events**	20,116 (16%)	41,355 (21%)
	**OR (95% CI)**	-	1.2 (1.2, 1.2)
	***p*-Value**	-	**<0.001**
**Length of hospital stay**	**Days**	9 (7–11)	10 (8–15)
	**Beta (95% CI)**	-	1.9 (1.9, 2)
	***p*-Value**	-	**<0.001**
**Costs**	**EUR**	7087 (6484–8122)	7400 (6484–10,492)
	**Beta (95% CI)**	-	1778 (1694, 1862)
	***p*-Value**	-	**<0.001**
**Pancreatitis**	**Events**	166 (0.1%)	636 (0.3%)
	**OR (95% CI)**	-	2 (1.8, 2.5)
	***p*-Value**	-	**<0.001**

## Data Availability

All data used in this work are stored in an anonymized fashion at the German Federal Statistical Office.

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
