# Peer review of "Surgical Trends and Complications in Partial and Radical Nephrectomy: Results from the GRAND Study"

_cancers, 2023, doi:10.3390/cancers16010097_

Round 1
Reviewer 1 Report
Comments and Suggestions for Authors
Congratulations on your efforts in completing this manuscript!
Partial nephrectomy respects an increased trend due mainly according to increasing the imagistic screening. The overall outcomes for this technique surpass the classical approach.
I have a few observations.
1. Is it possible to include the main operative technique - laparoscopic, robotic, or open?
2. Please reorganise Tables 2 and 3. It could be better to reverse data from rows and columns. It's hard to follow the information there.
3. Please extend the Discussion section. The topic is very interesting, and it allows plenty of comparisons.
Comments on the Quality of English LanguageMinor English issues.
Author Response
Congratulations on your efforts in completing this manuscript!
Partial nephrectomy respects an increased trend due mainly according to increasing the imagistic screening. The overall outcomes for this technique surpass the classical approach.
I have a few observations.
- Is it possible to include the main operative technique - laparoscopic, robotic, or open?
>>> We thank Reviewer 1 for the question. In Table 1, in the last three rows, the total number of cases performed with each operative technique (open, laparoscopic or robotic) are presented. Moreover, based on the volume of each center, the number of cases performed with each operative technique is also presented in Supplementary Material 1.
- Please reorganise Tables 2 and 3. It could be better to reverse data from rows and columns. It's hard to follow the information there.
>>> We thank Reviewer 1 for the proposal. Tables 2 and 3 are now reorganized to facilitate readers.
- Please extend the Discussion section. The topic is very interesting, and it allows plenty of comparisons.
>>> We thank Reviewer 1 for the suggestion. The Discussion section was now extended to approach the topic in a holistic approach based on the proposals of all Reviewers.
Reviewer 2 Report
Comments and Suggestions for Authors
The manuscript presents a comprehensive analysis of surgical trends and perioperative outcomes in renal cancer surgery using a large dataset from the GeRmAn Nationwide Inpatient Data (GRAND). The study is based on a substantial sample size (317,843 patients), making it one of the largest studies in the field. The manuscript is well organized, with clear headings and subheadings to help the reader find their way around. The selection criteria, including the categorization of centers based on annual case numbers, add granularity to the analysis. The inclusion of a wide range of variables, such as surgical approach, annual hospital case numbers and various perioperative outcomes, adds depth to the analysis. The results are presented in detail and the conclusions are supported by the evidence presented, highlighting the increasing use of partial nephrectomy and its associated benefits compared to radical nephrectomy.
Overall, I read the manuscript with great interest and have the impression that it is on trend and well-prepared for publication.
Consideration could be given to including a brief explanation of the clinical implications of the study results in the discussion section. Another crucial aspect to address is whether the GRAND dataset includes information on the decision-making process between partial and radical nephrectomy, including patient preferences. It would be valuable to explore the primary decision-making factors among patients with similar characteristics.
Author Response
The manuscript presents a comprehensive analysis of surgical trends and perioperative outcomes in renal cancer surgery using a large dataset from the GeRmAn Nationwide Inpatient Data (GRAND). The study is based on a substantial sample size (317,843 patients), making it one of the largest studies in the field. The manuscript is well organized, with clear headings and subheadings to help the reader find their way around. The selection criteria, including the categorization of centers based on annual case numbers, add granularity to the analysis. The inclusion of a wide range of variables, such as surgical approach, annual hospital case numbers and various perioperative outcomes, adds depth to the analysis. The results are presented in detail and the conclusions are supported by the evidence presented, highlighting the increasing use of partial nephrectomy and its associated benefits compared to radical nephrectomy.
Overall, I read the manuscript with great interest and have the impression that it is on trend and well-prepared for publication.
Consideration could be given to including a brief explanation of the clinical implications of the study results in the discussion section. Another crucial aspect to address is whether the GRAND dataset includes information on the decision-making process between partial and radical nephrectomy, including patient preferences. It would be valuable to explore the primary decision-making factors among patients with similar characteristics.
>>> We thank Reviewer 2 for the proposals. The Discussion section was now extended to focus on the clinical implications of the study results. Unfortunately, the GRAND dataset does not include information on the decision-making process between partial and radical nephrectomy, including patient preferences. This was added as an additional limitation of the present study.
Reviewer 3 Report
Comments and Suggestions for Authors
As recommended by urological societies, in the case of cT1 tumors partial nephrectomy (PN) should be preferred over radical nephrectomy (RN). The benefit of PN in terms of preserved renal function is still unclear, however. The American-Urological-Association (AUA) Guidelines for renal cancer (2017) recommend radical nephrectomy (RN) over partial nephrectomy (PN) whenever the oncologic risk is high stating that RN should be prioritized when all three other criteria are met: 1) increased tumor complexity; 2) no preexisting chronic-kidney-disease/proteinuria, and 3) normal contralateral kidney that will likely provide estimated glomerular-filtration-rate (eGFR) >45ml/min/1.73m2 even if RN is performed. However approximately one third of centres treating patients with renal cancer have acknowledged that the indication for PN versus RN does not correspond to these well-defined AUA Guidelines. Some of these considerations introduce important potential biases into your retrospective study.
My questions are:
1. Selection bias: Patients with larger or more advanced tumor stages are more likely to require radical nephrectomy, and this introduces a selection bias since obviously the outcomes reported may be influenced by inherent differences in tumor characteristics, the stage of the tumor and patients' general conditions rather than the surgical approach itself. Patients undergoing radical nephrectomy may have larger tumors or more advanced stages (TNM), which could significantly impact the estimated differences in terms of outcomes when comparing partial nephrectomy and radical nephrectomy. In the absence of detailed tumor characteristics, it is difficult to determine how an advanced tumor stage might affect outcomes.
2. Confounding Factors: The study adjusts for some major risk factors in the multivariate regression analysis. However, without specific tumor characteristics, there may be major residual confounding due to unmeasured variables related to tumor stage and extension.
3. Impact on generalizability: The findings may have limited generalizability, especially if the patient population undergoing radical nephrectomy differs systematically from those undergoing partial nephrectomy in terms of tumor characteristics and patients’ general conditions.
4. Clinical implications: The study's conclusions about the superiority of partial nephrectomy in terms of outcomes may need to be interpreted with caution, as the decision to perform radical nephrectomy might be driven by clinical factors related to an advanced tumor stage or tumor location and/or anatomical complexity.
5. Other data limitations: the study relies on retrospective administrative data, which may be prone to coding errors and misclassifications.
6. The study reports a significant increase in partial nephrectomy utilization over the years, with a 3-fold increase. In contrast, radical nephrectomy cases decreased by 40%. Does this observation indicate a decrease in patients with advanced tumor stages? In the absence of data, this issue should be discussed.
7. Hospital volume Impact: The proportion of partial to radical nephrectomies is higher in high-volume centers than in intermediate and low-volume centers. The study supports the trend towards centralization of complex urological operations, as higher annual hospital volume is associated with improved perioperative outcomes. How many non-urological surgical departments (general surgery) also perform renal cancer surgery?
8. Perioperative Outcomes: Radical nephrectomy is associated with higher odds of transfusion, sepsis, acute respiratory failure, acute kidney disease, and other complications than partial nephrectomy. Patients who undergo RN experience longer hospital stays and higher inpatient costs. Does the complications rate difference persist across low, intermediate, and high-volume centers?
9. Mortality rates: Radical nephrectomy is associated with 3.5 times higher odds of 30-day mortality than partial nephrectomy in the entire study population. Does the mortality difference persist for younger patients (< 60 years old)?
10. Surgical Approach Preferences: Minimal invasive renal surgery is preferred in only a small proportion of patients requiring nephrectomy in Germany, despite technological advancements. Robot-assisted nephrectomy is available in how many centers?
11. Comparative analysis of other healthcare systems. Try to find comparative studies across different healthcare systems in different countries. This could provide insights into the generalizability of the findings.
12. Impact of surgical approach on renal function. Considering the emphasis on functional outcomes after partial nephrectomy a deeper analysis of the impact of surgical approach (RN versus PN) on postoperative renal function could be really valuable.
Author Response
As recommended by urological societies, in the case of cT1 tumors partial nephrectomy (PN) should be preferred over radical nephrectomy (RN). The benefit of PN in terms of preserved renal function is still unclear, however. The American-Urological-Association (AUA) Guidelines for renal cancer (2017) recommend radical nephrectomy (RN) over partial nephrectomy (PN) whenever the oncologic risk is high stating that RN should be prioritized when all three other criteria are met: 1) increased tumor complexity; 2) no preexisting chronic-kidney-disease/proteinuria, and 3) normal contralateral kidney that will likely provide estimated glomerular-filtration-rate (eGFR) >45ml/min/1.73m2 even if RN is performed. However approximately one third of centres treating patients with renal cancer have acknowledged that the indication for PN versus RN does not correspond to these well-defined AUA Guidelines. Some of these considerations introduce important potential biases into your retrospective study.
My questions are:
- Selection bias: Patients with larger or more advanced tumor stages are more likely to require radical nephrectomy, and this introduces a selection bias since obviously the outcomes reported may be influenced by inherent differences in tumor characteristics, the stage of the tumor and patients' general conditions rather than the surgical approach itself. Patients undergoing radical nephrectomy may have larger tumors or more advanced stages (TNM), which could significantly impact the estimated differences in terms of outcomes when comparing partial nephrectomy and radical nephrectomy. In the absence of detailed tumor characteristics, it is difficult to determine how an advanced tumor stage might affect outcomes.
>>> We thank Reviewer 3 for highlighting an important limitation of our study. Indeed, in the GRAND study no information on histology is available. Nevertheless, the comment of Reviewer 3 was added in the third paragraph of the discussion section.
- Confounding Factors: The study adjusts for some major risk factors in the multivariate regression analysis. However, without specific tumor characteristics, there may be major residual confounding due to unmeasured variables related to tumor stage and extension.
>>> We thank Reviewer 3 for highlighting another important limitation of our study. Given that no information on histology is available, we could not adjust for tumor characteristics in the multivariable analysis. This was also added in the third paragraph of the discussion section.
- Impact on generalizability: The findings may have limited generalizability, especially if the patient population undergoing radical nephrectomy differs systematically from those undergoing partial nephrectomy in terms of tumor characteristics and patients’ general conditions.
>>> We thank Reviewer 3 for stressing that our findings may have limited generalizability. This was added as an additional limitation of the present study.
- Clinical implications: The study's conclusions about the superiority of partial nephrectomy in terms of outcomes may need to be interpreted with caution, as the decision to perform radical nephrectomy might be driven by clinical factors related to an advanced tumor stage or tumor location and/or anatomical complexity.
>>> We thank Reviewer 3 for providing insights into the clinical implications of our study. We have added this comment of Reviewer 3 as the last part of our conclusion.
- Other data limitations: the study relies on retrospective administrative data, which may be prone to coding errors and misclassifications.
>>> We thank Reviewer 3 for this comment. This has been already listed as the first and most important limitation of the present study.
- The study reports a significant increase in partial nephrectomy utilization over the years, with a 3-fold increase. In contrast, radical nephrectomy cases decreased by 40%. Does this observation indicate a decrease in patients with advanced tumor stages? In the absence of data, this issue should be discussed.
>>> We thank Reviewer 3 for bringing up this issue. The latter is now extensively discussion in the discussion.
- Hospital volume Impact: The proportion of partial to radical nephrectomies is higher in high-volume centers than in intermediate and low-volume centers. The study supports the trend towards centralization of complex urological operations, as higher annual hospital volume is associated with improved perioperative outcomes. How many non-urological surgical departments (general surgery) also perform renal cancer surgery?
>>> We thank Reviewer 3 for the valuable question. It was beyond the scope of the present study to assess the number of operations performed in urological versus non-urological surgical departments (e.g. general surgery, pediatric surgery). This was added as an additional limitation in the present study.
- Perioperative Outcomes: Radical nephrectomy is associated with higher odds of transfusion, sepsis, acute respiratory failure, acute kidney disease, and other complications than partial nephrectomy. Patients who undergo RN experience longer hospital stays and higher inpatient costs. Does the complications rate difference persist across low, intermediate, and high-volume centers?
>>> We thank Reviewer 3 for the question. In the separate multivariable analyses of patients undergoing renal surgery in low-, intermediate- and high-volume centers, radical nephrectomy was associated with worse perioperative outcomes compared to partial nephrectomy. The corresponding findings are presented in Supplementary Material 3.
- Mortality rates: Radical nephrectomy is associated with 3.5 times higher odds of 30-day mortality than partial nephrectomy in the entire study population. Does the mortality difference persist for younger patients (< 60 years old)?
>>> We thank Reviewer 3 for the question. All multivariate analyses were adjusted for age. Still, it was beyond the scope of the present analysis to explore differences in the perioperative outcomes across different patient groups (such as younger patients). This is now listed as a further limitation of the present study.
- Surgical Approach Preferences: Minimal invasive renal surgery is preferred in only a small proportion of patients requiring nephrectomy in Germany, despite technological advancements. Robot-assisted nephrectomy is available in how many centers?
>>> We thank Reviewer 3 for the question. To ensure anonymity, the Research Data Center excludes patient groups with fewer than three baseline characteristics or inpatient complications and does not allow comparisons on a hospital-level for any outcome. Therefore, this information cannot be retrieved. This was added in the Methods of the present study.
- Comparative analysis of other healthcare systems. Try to find comparative studies across different healthcare systems in different countries. This could provide insights into the generalizability of the findings.
>>> We thank Reviewer 3 for the suggestion. We have now added two further studies comparing radical with partial nephrectomy from Singapore and the Netherlands. Their main findings are now discussed in the second paragraph of the discussion.
- Impact of surgical approach on renal function. Considering the emphasis on functional outcomes after partial nephrectomy a deeper analysis of the impact of surgical approach (RN versus PN) on postoperative renal function could be really valuable.
>>> We thank Reviewer 3 for the proposal. As stated both in the Results and in the Discussion section of the manuscript, radical nephrectomy was associated with 60% higher odds for acute kidney disease. Accordingly, all multivariate models were adjusted for chronic renal disease. Nevertheless, both the perioperative laboratory values, as well as the degree of chronic kidney disease cannot be retrieved from the GRAND study. Furthermore, long-term outcomes are also not included in the present study. This was amended as a limitation of the present study.
Round 2
Reviewer 3 Report
Comments and Suggestions for Authors
All reviewers' questions and criticisms were satisfactorily answered by the Authors.